# Global modeling of brown carbon: impact of temperature- and humidity-dependent bleaching

Xinchun Xie<sup>1,2,3</sup>, Yuzhong Zhang<sup>2,3\*</sup>, Ruosi Liang<sup>1,2,3</sup>, Xuan Wang<sup>4</sup>

¹ College of Environmental and Resource Sciences, Zhejiang University, Hangzhou, Zhejiang 310058,
China

<sup>2</sup> Key Laboratory of Coastal Environment and Resources of Zhejiang Province, School of Engineering, Westlake University, Hangzhou, Zhejiang 310030, China

<sup>3</sup> Institute of Advanced Technology, Westlake Institute for Advanced Study, Hangzhou, Zhejiang 310024,

China

<sup>4</sup> School of Energy and Environment, City University of Hong Kong, Kowloon Tong 999077 Hong Kong SAR, China;

Correspondence to: Yuzhong Zhang (zhangyuzhong@westlake.edu.cn)

Abstract: Brown carbon (BrC), a light-absorbing component of organic aerosols, undergoes bleaching in the atmosphere, a process where its light-absorption capacity diminishes over time due to chemical transformation. A recent study suggests that the lifetime of freshly emitted, unbleached BrC (referred to as fresh BrC) against bleaching (τ<sub>BrC</sub>) is influenced by ambient temperature and relative humidity. In this study, we incorporate the improved  $\tau_{BrC}$  parameterization into the atmospheric chemical transport model (GEOS-Chem) to assess its atmospheric chemical and radiative effects. Our results show that  $\tau_{BrC}$  varies strongly with altitude, ranging from 1–10 hours in the planetary boundary layer (PBL) to over 100 hours in the upper troposphere, where bleaching becomes negligible. Dry regions (e.g., Northern Africa and South Asia) exhibit longer surface  $\tau_{BrC}$ , while humid regions (e.g., tropics) show shorter  $\tau_{BrC}$ . The updated  $\tau_{BrC}$  parameterization triples the global burden of fresh BrC compared to the baseline parameterization with uniform  $\tau_{BrC}$ , increasing its effective lifetime from 0.45 to 1.45 days and amplifying the direct radiative effect (DRE) of BrC by 48% (from +0.059 to +0.088 W m<sup>-2</sup>). Lofted wildfire emissions experience prolonged  $\tau_{BrC}$  due to reduced bleaching in the free troposphere, underscoring the importance of fire injection height. Additionally, BrC absorption suppresses photochemical activity, reducing the photolysis rate of NO<sub>2</sub> (JNO<sub>2</sub>) by up to 7.4%, surface ozone by 0–2.5% and tropospheric OH by 0-6.9%. These effects intensify during major wildfire events, such as Siberian fires in 2019 that caused JNO<sub>2</sub> and ozone to drop by 36.3% and 17.5%, respectively, highlighting BrC's role in perturbing atmospheric oxidation capacity.

**Keywords:** brown carbon, bleaching lifetime, biomass burning, direct radiative effect, atmospheric chemistry

#### 1. Introduction

Brown carbon (BrC), a highly complex and dynamic mixture of organic aerosols (Lin et al., 2017), can absorb sunlight in the ultraviolet (UV) and visible wavelengths, contributing to the warming of the atmosphere (Washenfelder et al., 2015; Andreae and Gelencsér, 2006; Laskin et al., 2015). Previous studies indicate that BrC accounts for 7–19% of total aerosol light absorption (Feng et al., 2013), with its influence reaching up to 40–50% at 300–400 nm wavelengths (Lack and Langridge, 2013). Studies have reported that the atmospheric radiative forcing due to BrC can be as much as 40% of that caused by black carbon (BC) in regions with intensive agricultural fires (Srinivas et al., 2016), and approximately 24% in the free troposphere affected by biomass burning (Zhang et al., 2017).

Biomass burning emissions are the primary source of global BrC (Andreae and Merlet, 2001; Andreae and Gelencsér, 2006), contributing ~60% of BrC warming effect (Yue et al., 2022), followed by contributions from biofuel burning (e.g., residential wood combustion) (Saleh et al., 2014). Frequent and intensive wildfire events, driven by climate change, land-use change, and natural variability (Fu et al., 2013; Jolly et al., 2015), have raised the concerns of positive feedback through wildfires (Zheng B et al., 2023; Cunningham et al., 2024). Large-scale fires emit substantial amounts of aerosols, disrupting the Earth's radiation balance (Huang et al., 2023) and adversely affecting human health (Nie et al., 2018). Intensified fire activity has slowed down or even reversed the downward trend in global and regional PM<sub>2.5</sub> levels (Li et al., 2023; Burke et al., 2023).

Most climate models treat BC as the only significant light-absorbing aerosol, neglecting the contribution of BrC in the near-UV to visible range (Kelesidis et al., 2022). However, recent studies have highlighted the contribution of BrC to atmospheric radiative forcing. Collow et al. (2024) found that including BrC significantly improved simulated absorption in the ultraviolet band. Zhang et al. (2020) estimated that the direct forcing

effect (DRE) from BrC is around +0.1 W m<sup>-2</sup>, corresponding to about 25% of the BC (0.39 W m<sup>-2</sup>). Feng et al. (2013) demonstrated that the radiative effect of organic aerosols in certain regions of the tropopause could shift from cooling (-0.08 W m<sup>-2</sup>) to warming (+0.025 W m<sup>-2</sup>) when BrC is included in radiative simulations, a result consistent with Delessio et al. (2024), who reported a BrC DRE of 0.04 W m<sup>-2</sup>. Srinivas et al. (2016) found that BrC accounts for 40% of the atmospheric radiative forcing caused by BC, and Chung et al. (2012) estimated that 20% of the solar radiation absorption by carbonaceous aerosols at 550 nm originates from BrC. When BrC bleaching is considered, the reported global DRE values are reduced to 0.06 W m<sup>-2</sup> (Brown et al., 2018) and 0.029 W m<sup>-2</sup> (Drugé et al., 2022). This warming effect of carbonaceous aerosols in the troposphere is especially evident in regions with significant biomass burning.

BrC primarily absorbs light in the near UV-visible spectrum, reducing sunlight availability for photolysis reactions. This absorption alters the local photochemical environment, reducing the photolysis of nitrogen dioxide (NO<sub>2</sub>) to nitric oxide (NO) - a crucial step in ozone formation, leading to lower ozone levels in regions with high BrC concentrations. Other free radicals (e.g., OH and HO2) are similarly affected (Wu et al., 2019). OH radicals, which are key atmospheric oxidants, are primarily generated through the reaction between O(¹D) (produced by UV-induced ozone decomposition) and water molecules. Studies have shown that BrC emissions from biomass burning can reduce OH concentrations by 5–30% in affected areas (Hammer et al., 2016). While the effects of BrC on atmospheric oxidation are currently underexplored (Selimovic et al., 2020), further investigation is essential to understand its chemical implications.

Atmospheric BrC evolve through photochemical aging, where exposure to sunlight and atmospheric oxidants modifies its chemical structure and optical properties. Over short timescales, BrC can darken near its source, while bleaching and oxidative whitening occur over longer timescales (e.g., a day or more after emission). The bleaching time

(referred to as lifetime in this study) of BrC is a critical factor that influences its climatic impact, but it remains challenging to quantify accurately. Accurately parameterizing the lifetime of BrC in climate models is essential for predicting its distribution, concentration, and radiative effects. Most studies simplify BrC's lifetime to either exclude bleaching or assume a constant value of about one day (Feng et al., 2013; Jo et al., 2016; Wang et al., 2018). However, wildfire-driven convective vertical transport can extend BrC's lifetime, as particle viscosity affects its atmospheric persistence (Schnitzler et al., 2022; Xu et al., 2024). The lifetime in the upper troposphere is significantly longer than one day, and there are also differences in different dimensional bands (Schnitzler et al., 2022). Neglecting these variations leads to inaccurate estimates of BrC's effects in models.

In this study, we update the bleaching lifetime parameterization of BrC in the GEOS-Chem atmospheric chemical transport model, and examine the impacts of this update on the radiative effects of wildfire-derived BrC. Unlike BC, whose climate impacts are relatively well understood, BrC's atmospheric warming effect remains highly uncertain and poorly represented in climate models. Given the expected increase in wildfire activity globally, quantifying BrC's radiative impacts is becoming increasingly critical. This study investigates the influence of wildfire emissions on the distribution of atmospheric BrC and systematically evaluates its climate and chemical impacts. We first update the parameterization of BrC's chemical lifetime against bleaching and examine its environmental dependence (Section 3.1). We then explore the global distribution of BrC (Section 3.2), and evaluate the simulation results against multiple ground-based and aircraft observations (Section 3.3). Finally, we quantify the direct radiative effect (DRE) of BrC at regional and global scales (Section 3.4) and assess its influences on atmospheric oxidation capacity and chemical processes (Section 3.5).

#### 2. Method

#### 2.1 GEOS-Chem simulation

The global simulations utilize version 12.8 of GEOS-Chem with a horizontal resolution of 4 ° × 5 ° and 47 vertical levels, which is driven by meteorology fields from the Modern Era Retrospective-analysis for Research and Applications, version 2 (MERRA-2) (Gelaro et al., 2017) (https://gmao.gsfc.nasa.gov/reanalysis/MERRA-2/, Last access: 20 February, 2024). Global anthropogenic emissions are from EDGARv43 (Crippa et al., 2018), with region-specific updates including APEI for Canada (Canada, 2016), NEI (2014) for the USA, MIX for Asia (Li et al., 2017), and DICE\_Africa for Africa (Marais and Wiedinmyer, 2016). Biofuel emissions are from Bond et al. (2007), and daily resolved biomass burning emissions from the Global Fire Emissions Database with small fires (GFED4s) (Giglio et al., 2013; Van Der Werf et al., 2017; Randerson et al., 2017). These emission inventories yield global OC emissions of 42.74 Tg and BC emissions of 8.24 Tg in 2019. Of the total OC emissions, 21.17 Tg are from biomass burning and 6.29 Tg from biofuel burning. In the simulation, these sources are treated as BrC (see Section 2.2 for details), while the remaining OC emissions are modeled as purely scattering aerosols.

In the simulation, 80% of BC, 50% of anthropogenic OC, and 50% of biomass burning OC are emitted as hydrophobic. These hydrophobic aerosols are converted to hydrophilic forms, which are subject to wet removal, with an e-folding timescale of 1.2 days (Park and Jacobb, 2003). BC and OA are treated as externally mixed as described in Wang et al. (2014), with absorption enhancement of 1.1 for fossil BC and 1.5 for biofuel and biomass burning BC. POA is diagnosed from simulated primary OC using an OA/OC mass ratio of 2.1 (Wang et al., 2018). SOA aerosols are simulated using the simple scheme (Kim et al., 2015; Pye et al., 2010), with all SOA assumed hydrophilic. Inorganic sulfate-nitrate-ammonium aerosols are simulated following (Shah et al., 2020; Shao et al., 2019), with thermodynamic equilibrium computed using the ISORROPIA II (Fountoukis and Nenes, 2007). Dust and sea salt aerosols are simulated according to Duncan Fairlie et al. (2007) and Jaeglé et al. (2011), respectively.

To evaluate the radiative impact of BrC, we use the Rapid Radiative Transfer Model for Global Circulation Models (RRTMG), coupled with GEOS-Chem, to calculate both longwave and shortwave radiative fluxes and heating rates in the atmosphere. RRTMG is invoked every three hours and the computation of instantaneous atmospheric radiative fluxes is performed at 30 wavelengths between 0.23 to  $56 \mu m$  in the clear sky (Heald et al., 2014).

#### 2.2 Brown carbon simulation

Our simulation treats organic aerosols from biomass or biofuel burning as light-absorbing BrC and those from fossil fuel combustion as purely scattering. Although a few field studies, particularly in China, have reported weak light absorption by primary organic aerosols from fossil fuel sources (Yan et al., 2017; Chen et al., 2020), their overall contribution is minor on the global scale and thus is not considered in our simulation.

To capture the change of mass absorption efficiency (MAE) due to chemical processing, organic aerosols from biomass and biofuel burning are modeled as two distinct species: fresh BrC, which is strongly absorbing, and bleached BrC, which is weakly absorbing, following the approach by Wang et al. (2018). All freshly emitted OA from biomass and biofuel burning are specified as fresh BrC. They are subsequently converted to bleached BrC in the atmosphere, with the rate governed by bleaching lifetime  $\tau_{BrC}$ .

The BrC optical properties are implemented following Wang et al. (2018). The MAEs of fresh BrC at different wavelengths are estimated based on the Mie theory as a function of aerosol size, aerosol density, and the refractive index. The size of OA is assumed to follow a log-normal distribution, with a geometric median diameter of 180 nm and a standard deviation of 1.6 (Wang et al., 2018). The density of OA is set at 1.3 g cm<sup>-3</sup>. The real part of the refractive index is 1.5, while the imaginary part and its wavelength dependence are parameterized based on the mass emission ratios of BC/OA (June et al., 2020). The BC/OA ratios are specified as 0.05 for biomass burning and

0.12 for biofuel, representing average global burning conditions (Wang et al., 2018).

Based on these parameters, Mie calculations yield an MAE of 1.3 m<sup>2</sup> g<sup>-1</sup> at the 365 nm 205 wavelength for fresh BrC from biomass burning and 1.2 m<sup>2</sup> g<sup>-1</sup> for biofuel burning, with corresponding absorption Ångström exponents (AAE) between 300 and 600 nm of 3.1 and 2.6, respectively. To account for weakened absorption by bleaching, we assume that bleached BrC absorbs only one-quarter as much as fresh BrC, resulting in MAEs of 0.37 m<sup>2</sup> g<sup>-1</sup> and 0.30 m<sup>2</sup> g<sup>-1</sup> for biomass and biofuel burning aerosols, respectively, at 210 365 nm.

The conversion from fresh to bleached BrC is governed by the chemical lifetime  $\tau_{\rm BrC}$ (referred to as bleaching lifetime hereafter). Wang et al. (2018) parameterized this parameter as inversely proportional to the ambient hydroxyl radical (OH) concentration, specifying  $\tau_{BrC}$  as  $\frac{5\times10^5}{[OH]}$  d where [OH] is the OH concentration in molec·cm<sup>-3</sup>. This formulation, mainly based on empirical evidence from field measurements in the lower troposphere (Forrister et al., 2015; Wang et al., 2018), yields roughly a  $\tau_{\rm BrC}$  of about one day throughout the troposphere.

215

More recent laboratory experiments by Schnitzler et al. (2022) found that the chemical lifetime of water-soluble BrC, generated from smoldering pine wood in the presence of ozone, increases substantially with enhanced particle viscosity when temperature and RH get lower. Although the experimental setup (i.e., using ozone as the oxidant, generating aerosols from pine wood, and focusing on water-soluble BrC) does not fully represent atmospheric conditions, the finding that  $\tau_{BrC}$  is highly sensitive to environmental parameters has critical implications for BrC's radiative and chemical effects.

To quantitatively evaluate these implications, we implement the parameterization of  $\tau_{BrC}$  (in s) developed by Schnitzler et al. (2022) in GEOS-Chem. As shown in Eq. (1),

 $\tau_{\rm BrC}$  depends on ambient temperature (T) and relative humidity (RH):

$$\tau_{\rm BrC} = A \left[ D_0(T) \left( \frac{\eta_0(T)}{\eta_{\rm BrC}(T,RH)} \right)^{\xi} \right]^{-\frac{1}{2}} \tag{1}$$

where  $D_0$  is the diffusion rate of ozone in pure water, calculated with the Stokes-Einstein equation as a function of T.  $\eta_0$  is the viscosity of pure water as a function of T, and  $\eta_{BrC}$  is the viscosity of aqueous BrC aerosols, parameterized as a function of T and RH based on the experimental measurements (Schnitzler et al., 2022). A and  $\xi$  are treated as constants where A=0.125 m s<sup>-1/2</sup> and  $\xi$ =0.684.  $\tau_{BrC}$  is calculated and updated online in the GEOS-Chem simulation driven by T and RH data from MERRA-2.

235

## 2.3 Observational data

We use observations from aircraft campaigns (ATom mission, Figure S1) and a groundbased remote sensing network (AERONET) to evaluate our simulations of BrC. The Atmospheric Tomography Mission (ATom) is a NASA-led aircraft campaign aimed at measuring and understanding the global distribution of greenhouse gases and reactive gases and aerosols (Thompson et al., 2022). The ATom flights, conducted between 2016 and 2018, sampled extensive latitudinal and longitudinal ranges, from the Arctic to the Antarctic and over both the Pacific and Atlantic Oceans (Prather et al., 2017). During the ATom -2, ATom -3, and ATom -4, BrC was sampled at altitudes ranging from 0.2 to 12 km aboard the NASA DC-8 aircraft, alongside comprehensive measurements of various gases and aerosols. Water-soluble BrC (WS BrC) in aerosols was measured with two systems: a spray chamber (MC) coupled to a liquid waveguide capillary cell (LWCC), and a particle-into-liquid sampler (NOAA PILS-LWCC) coupled to an LWCC (Washenfelder et al., 2015). The MC-LWCC system collects aerosols through a spray chamber, trapping the particles in water, which are then analyzed using an LWCC and a spectrometer. The PILS-LWCC system collects aerosols into a liquid stream for similar analysis. The detection limits at 365 nm were 1.53 Mm<sup>-1</sup> (MC), 0.89 Mm<sup>-1</sup> (CSU), and 0.03 Mm<sup>-1</sup> (NOAA), respectively. A factor of 2 is used to convert the light absorption of BrC in solution to the light absorption of atmospheric particles (Liu et al., 2013; Zhang et al., 2017). In this study, we evaluate

our simulations against the measurements taken during ATom -4, which occurred from April 24 to May 21, 2018 (https://espoarchive.nasa.gov/archive/browse/atom/id14/DC8, Last access: 25 October, 2024). Samples were collected every 5 minutes at altitudes below 3 km and every 15 minutes at higher altitudes, yielding a total of 1,074 samples. The flight path in ATom-4 is shown in Figure S1.

The AERONET (AErosol RObotic NETwork) program is a federation of ground-based remote sensing aerosol networks established by NASA and LOA-PHOTONS (CNRS) (Holben et al., 2006). The program provides a long-term, continuous and readily accessible public domain database of aerosol optical, microphysical and radiative properties. AERONET's aerosol optical depth (AOD) observations are derived from direct solar radiation in multiple wavelength bands, mainly ranging from 340 nm to 1640 nm, while other aerosol properties (including single scattering albedo SSA and absorbing aerosol optical depth AAOD) are derived from diffuse sky radiation at four wavelengths: 440, 675, 870 and 1020 nm (Dubovik and King, 2000). This study extracted AOD and AAOD at 440 nm from AERONET data in 2019 to compare with the model simulation results. We use high-quality Level 2.0 AOD data (uncertainty of 0.01 in the visible range and 0.02 in the ultraviolet range (Giles et al., 2019) and almucantar inversion AAOD data. To ensure the consistency in the comparison between model simulations and observations, we sample the simulation on days and locations when AOD and AAOD measurements are available.

## 2.4 Simulation experiments

We perform a series of global  $4^{\circ}\times 5^{\circ}$  GEOS-Chem simulations for 2019 to examine the impact of environmental-condition-dependent BrC chemical lifetime ( $\tau_{\rm BrC}$ ) on the abundance and distribution of BrC globally (Table 1). The baseline simulations (Base and BaFt) use Wang et al. (2018)  $\tau_{\rm BrC}$  parameterization (bleaching lifetime of roughly 1 day, largely invariant with environmental conditions). The updated simulations (Upd

and UpdFt) incorporate the Schnitzler et al.  $(2022) \tau_{BrC}$  parameterization as a function of temperature and RH (Eq.1). To test the impact of vertical partitioning of fire emissions and its interactions with  $\tau_{BrC}$  parameterizations, we perform GEOS-Chem simulations that assign 100% (0%) (Base and Upd) and 65% (35%) (BaFt and UpdFt) of wildfire emissions (all fire-emitted species including BC and BrC) in the boundary layer (free troposphere). The 65%:35% partitioning is based on the averages of aerosol smoke plume heights observed by the Multi-angle Imaging SpectroRadiometer (MISR) (Val Martin et al., 2010) and has been previously applied in GEOS-Chem modeling studies to assess the impact of fire plume heights (Fischer et al., 2014; Jin et al., 2023). In addition to 2019, we also run the April and May 2018 simulation to evaluate against the ATom-4 observations. We also use the GFED5 beta version to test the sensitivity to biomass emission inventories.

Table 1. descriptions of model sensitivity experiments.

| Experiments | Lifetime       | Emission inventory | Fire emissions fraction under PBL |
|-------------|----------------|--------------------|-----------------------------------|
| Base        | original τ_BrC | GFED4s             | 100%                              |
| BaFt        | original τ_BrC | GFED4s             | 65%                               |
| Upd         | Updated τ_BrC  | GFED4s             | 100%                              |
| UpdFt       | Updated τ_BrC  | GFED4s             | 65%                               |
| UpdGF5      | Updated τ_BrC  | GFED5              | 100%                              |

#### 3. Results and Discussion

# 3.1 Chemical lifetime of BrC against bleaching

Figure 1 shows the spatial distribution of  $\tau_{BrC}$  as parameterized based on Schnitzler et al. (2022). In the planetary boundary layer (

Figure 1. The lifetime of BrC as a function of altitude and latitude (a) and as a function of temperature and relative humidity (b).

Figure 2. Global distribution of fresh BrC column density (a), total BrC (fresh BrC + bleached BrC) column density (c), and fresh to total BrC ratio (e) with updated  $\tau$ \_BrC (UpdFt). Difference in the global distribution of BrC column density (b), total BrC (fresh BrC + bleached BrC) column density (d), and fresh to total BrC ratio (f) between updated  $\tau$  BrC and original  $\tau$  BrC simulations (UpdFt and BaFt).

Figure 3. Vertical profiles (a, b) and column densities (c) of fresh BrC concentrations of global, and in the Northern Hemisphere Extratropics, Tropics, and Southern Hemisphere Extratropics simulated by the GEOS-Chem model.

## 3.2 Global distribution of fresh BrC

Figure 2a shows the global distribution of fresh BrC column density in 2019, which reveals elevated concentrations in regions characterized by substantial biomass burning and biofuel emissions (Figure S3). These hotspots include tropical Africa (biomass), tropical South America (biomass), East Siberia (biomass), South Asia (biofuel), and East China (biofuel). Compared to the baseline simulation (BaFt), the updated simulation (UpdFt) with temperature- and RH-dependent  $\tau_{BrC}$  yields higher fresh BrC concentrations across most regions. The largest increase is observed in southern Africa, which is due to long  $\tau_{BrC}$  near the surface (Figure 1) resulting from the region's strong biomass burning emissions (Figure S3) and dry environmental conditions (Figure S2). Conversely, despite intensive wildfire emissions in East Siberia, the updated simulation

shows minimal differences in fresh BrC concentrations because of comparable  $\tau_{\rm BrC}$  values between the two parameterizations (Figure S2a).

Figure 3 compares the vertical profiles and average column densities of fresh BrC concentrations across simulations. The updated simulation (UpdFt), with revised  $\tau_{BrC}$  and fire injection parameterizations, yields the highest fresh BrC concentrations, with average column densities of 235 µg m<sup>-2</sup> in the tropics, 172 µg m<sup>-2</sup> in the Northern Extratropics, and 24 µg m<sup>-2</sup> in the Southern Extratropics. These values represent factors of 4.7, 1.6, 8.6 increases, respectively, compared to the Base simulation. Globally, the updated simulation (UpdFt) produces an average column density of 144.4 µg m<sup>-2</sup>, about three times that of the baseline simulation (Base, 53.0 µg m<sup>-2</sup>). Given the global average emission fluxes of BrC, this result translates to an effective  $\tau_{BrC}$  of 1.45 days in the UpdFt simulation, as compared to 0.45 days in the Base simulation. Comparison with BaFt and Upd simulations (average column density of 52.9 µg m<sup>-2</sup> and 120.7 µg m<sup>-2</sup>) suggests that the increase in BrC abundance in UpdFt relative to Base is mainly due to the bleaching lifetime update.

Vertical profiles of fresh BrC demonstrate significant differences between the UpdFt and Base simulations. The absolute differences are particularly large in the lower troposphere of the tropics, reaching up to  $0.05~\mu g~m^{-3}$ , while relative differences are most pronounced in the upper troposphere, exceeding a factor of 10 despite lower absolute concentrations. Meanwhile, the updated  $\tau_{BrC}$  parameterization does not change the total BrC concentrations (fresh + bleached BrC) (Figure S4), as the abundance of total BrC is controlled by the rate of wet and dry deposition.

The influence of  $\tau_{BrC}$  parameterization on fresh BrC concentrations is modulated by fire emission injection height. Direct injection of emissions into the free troposphere results in slower BrC bleaching and higher concentrations at elevated altitudes. Comparing results from UpdFt and Upd simulations, we find that under environmental-

condition-dependent  $\tau_{BrC}$  parameterization, injecting 35% of fire emissions into the free troposphere (UpdFt) (Thapa et al., 2022) leads to 30-50% greater fresh BrC burdens compared to those with emissions confined to the surface layer (Upd) (Figure 3). In contrast, under original uniform  $\tau_{BrC}$ , different treatments of fire emission injections (BaFt and Base) affect only the vertical distribution of BrC, without influencing its overall atmospheric burden (Figure 3). These results underscore the interactions between  $\tau_{BrC}$  and fire emission injection heights.

## 3.3 Evaluation against observations

We compare our simulations with the aircraft campaign ATom-4 which measured the global distribution of BrC over remote oceans (Figure S1) (Zeng et al., 2020). The baseline simulation severely underestimates BrC absorption at 365 nm (Abs<sub>365</sub>) (Figure 4a). The updated simulation with longer  $\tau_{\rm BrC}$  increases fresh BrC concentrations and hence their Abs<sub>365</sub> by about an order of magnitude, bringing the simulation results closer to but still lower than the observations (Figure 4a). Meanwhile, the model generally captures observed levels of OA and CO during ATom-4, but with an underestimation of OA in the upper troposphere and an overestimation of CO in the lower troposphere (Figure 4).

Figure 4. Vertical distribution of the median light absorption coefficient of BrC at 365 nm (a), total OA concentration (b) and  $\Delta$ CO concentration (c) from simulation and observations (ATom-4).  $\Delta$ CO is calculated as the CO concentration minus the background value (0.064 mg m<sup>-3</sup> for observations, 0.0194 mg m<sup>-3</sup> for UpdFt, and 0.0189 mg m<sup>-3</sup> for UpdGF5).

Figure 5 and Figure 6 shows the comparison of the annual means of AOD and AAOD simulation results against the AERONET observations. The model in general well reproduces both the magnitude and the distribution of observed AOD. The model also captures high AAOD observed in southern and eastern Asia, where the absorption is mainly due to BC from anthropogenic sources, but underestimates high AAOD observed in regions with strong influence of biomass burning, such as Africa, South America, and Siberia (Figure 5). Both the baseline and updated simulation show good agreement with observed AOD (R = 0.88, slope = 0.79 in BaFt and R = 0.89, slope = 0.80 in UpdFt). The updated UpdFt simulation (R = 0.56, slope = 0.42) improves agreement with AAOD observations relative to the BaFt simulation (R = 0.47, slope = 0.30), but still has notable underestimation (Figure 6). Meanwhile, at some stations with low SSA values, the simulated SSA is overestimated (Figure S5).

The underestimation of Abs<sub>365</sub> against ATom-4 data (Figure 4) and AAOD against AERONET data (Figure 5 and 6) may partly be explained by underestimation of biomass burning emissions. We perform additional simulations with the newly released GFED5 fire emission inventory (https://globalfiredata.org/, Last access: 10 April, 2025), which generally predicts higher biomass burning emissions than the GFED4s inventory. This simulation leads to better alignment with both ATOM-4 Abs<sub>365</sub> (Figure 4) and AERONET AAOD observations (Figure 6). However, the GFED5 simulatoin also leads to an overestimation of OA in the lower troposphere against ATom-4 data and AOD against AERONET data (Figure 4 and 6).

Additionally, there are also considerable uncertainties in the BrC simulation associated with its optical properties. The MAE applied in different modeling studies vary considerably (Zhang et al., 2020; Jo et al., 2016), and laboratory measurements have also demonstrated source- and season-dependent differences in MAE (Chen et al., 2018; Xie et al., 2020). Moreover, assumptions about particle size distribution and refractive

index also contribute to the uncertainties and need to be further constrained using observational and experimental data (Wu et al., 2020; Shamjad et al., 2018). Future improvement of BrC simulations may including refined treatment of these factors.

Figure 5. Global annual mean distribution of AOD (a, c) and AAOD (b, d) at 440 nm with updated  $\tau_{BrC}$  using GFED4s (a, b) and GFED5 (c, d) emission inventories. The dots are the observed annual AOD and AAOD from AERONET at 440 nm.

Figure 6. Comparison between observed and simulated AOD (a) and AAOD (b).

Observations are from AERONET.

# 3.4 Direct radiative effect of BrC

The presence of BrC enhances the overall absorptivity of aerosols, leading to positive direct radiative effects (DRE) at the top of the atmosphere. Incorporating an improved

 $\tau_{BrC}$  parameterization into the model increases the DRE of BrC absorption from +0.059 W m<sup>-2</sup> in the baseline simulation to +0.088 W m<sup>-2</sup> in the UpdFT simulation (Figure 7). This 48% enhancement in DRE highlights the sensitivity of BrC's radiative effects to the rate of bleaching. Our estimate of the BrC DRE is within the estimates by previous studies (0.04 – 0.11 W m<sup>-2</sup>) (Feng et al., 2013; Zhang et al., 2020; Wang et al., 2018). The value is higher than the estimate by Wang et al. (2018) (+0.048 W m<sup>-2</sup>), which did not account for longer  $\tau_{BrC}$  in the cold and dry environment, but is lower than those reported by Zhang et al. (2020) (+0.10 W m<sup>-2</sup>), which made an assumption that convection-lofted BrC does not photo-bleach. The light absorption of BrC reduces the negative DRE of OA by 5.7% in the Base simulation and by 8.4% in the UpdFT simulation (Figure 7).

Figure 7. The DRE of original  $\tau$ \_BrC (a) and updated  $\tau$ \_BrC (b), and pure scattering OC (c).

# 3.5 Effects of BrC on atmospheric chemistry

BrC reduces the ultraviolet light available for photochemical reactions, affecting atmospheric chemistry. Figure 8 evaluates the effect of BrC light absorption on the photolysis rate of  $NO_2$  (JNO<sub>2</sub>) and concentrations of  $O_3$  and OH radicals. The effects are most pronounced in regions with substantial biomass burning emissions. The annual average of JNO<sub>2</sub> decreases by 0-7.4%, while surface ozone levels decline by 0-2.5%. These results are consistent with, but slightly lower than the results of Jo et al. (2016). The light absorption of BrC also results in a 0-6.9% reduction in tropospheric OH concentration, which is also lower than those reported in the Northern Hemisphere in Jo et al. (2016) (0-10%) and Jiang et al. (2012) (up to 15%).

The effects of BrC on atmospheric chemistry are more substantial in the season and region with intensive biomass burning. For instance, the reduction in JNO<sub>2</sub> and surface ozone concentration due to BrC reached 36.3% and 4 ppb (17.5%), respectively, in high latitude Siberia, Russia in August 2019 (Figure S6), when a large wildfire event occurred (Konovalov et al., 2021a; Konovalov et al., 2021b). Another example is the large wildfire over tropical Malaysia in September 2019, which accounted for approximately 40% of its annual total wildfire emissions. In this case, JNO<sub>2</sub> decreased by about 27.6%, and surface O<sub>3</sub> concentrations by as much as 6.4 ppb (10.8%) (Figure S7).

Figure S8 shows the overall responses of atmospheric chemistry to wildfire emissions including reactive gases and aerosols. Wildfire emissions increase tropospheric ozone concentrations across the globe, while increasing OH concentrations near fire sources and suppressing OH concentrations away from the source. Comparison of Figure S8 and Figure 8 shows that including BrC light absorption partially offsets the ozone enhancement from wildfire emissions. Moreover, BrC absorption reduces near-source OH enhancement caused by fires while reinforcing OH reduction globally. These results demonstrate that neglecting BrC light absorption in models leads to moderate biases in simulated chemical responses to wildfire emissions.

Figure 8. The effect of BrC absorption on global surface photolysis rates JNO<sub>2</sub> (a), O<sub>3</sub> (b), and column OH (c).

## 4. Discussion

In this study, we adopt the bleaching lifetime parameterization derived from Schnitzler et al. (2022), who demonstrated that BrC bleaching rates are governed by aerosol viscosity, which itself depends strongly on temperature and relative humidity (RH). This environmental dependence introduces significant regional and vertical variability in BrC transformation, in marked contrast to rather uniform bleaching lifetime of  $\sim 1$  day applied in previous modeling studies.

525

530

520

The finding by Schnitzler et al. (2022) is consistent with observations by Zhang et al. (2017), who identified substantial presence of BrC at high altitudes over regions affected by biomass burning. An earlier modeling study (Zhang et al., 2020) attempted to reconcile such observations with short bleaching lifetimes by assuming infinite lifetimes to convection-lofted aerosols, albeit without mechanistic justification.

Schnitzler et al. (2022) provide this missing mechanistic basis.

While representing an important step forward, several aspects of the Schnitzler et al. (2022) parameterization merit further investigation. First, the experiment examined only ozone-driven bleaching. The role of other oxidants such as OH and their sensitivity to environmental parameters are yet to be explored. Second, the study focused on water-soluble BrC, though field evidence indicates that water-insoluble BrC accounts for a large fraction of BrC absorption. The impact of temperature and RH on bleaching behaviors of water-insoluble BrC remains uncharacterized. Third, the parameterization derives from BrC produced by smoldering pinewood, while real-world BrC properties may vary considerably across different fuel types and combustion conditions (Sun et al., 2021; Cai et al., 2023). Finally, neither the current study nor Schnitzler's work addresses secondary BrC formation pathways, which may involve different bleaching mechanisms. These limitations highlight important directions for future research to further refine the representation of BrC transformation in atmospheric models.

#### 5. Conclusions

This study implements an updated  $\tau_{BrC}$  parameterization developed by Schnitzler et al. (2022) in the GEOS-Chem model and evaluates its impact on BrC's radiative and chemical effects.  $\tau_{BrC}$  is parameterized to strongly depend on temperature and relative humidity. This results in  $\tau_{BrC}$  that varies by orders of magnitude, from just a few hours in warm, humid boundary layers to over 2 weeks in cold, dry upper atmospheres. In contrast, the original parameterization in the model is a weak function of OH concentration and thus rather uniform throughout the troposphere.

555

Compared to the original parameterization, the revised  $\tau_{BrC}$  increases the global average BrC lifetime from 0.45 to 1.45 days, leading to a tripling of global fresh BrC burdens. The revised  $\tau_{BrC}$  also leads to higher BrC concentrations in the middle and upper troposphere, particularly evident over source regions with deep convective activity,

such as Central Africa and South Asia. In addition, injecting fire emissions into the free troposphere increases fresh BrC burdens by 30-50% relative to surface-only emissions, highlighting the interplay between fire injection and BrC persistence.

The updated  $\tau_{BrC}$  parameterization improves the agreement between simulation and observations, particularly in the upper troposphere, but still generally underestimates BrC absorption (Abs<sub>365</sub>) in the ATom-4 campaign and AAOD in AERONET biomass burning regions. Using GFED5 fire emission inventory, which has higher wildfire emissions, instead of GFED4s, can address these underestimations, but leads to overestimations in OA and AOD.

560

565

The revised bleaching lifetimes lead to a 48% increase in BrC's global direct radiative effect from +0.059 to +0.088 W m<sup>-2</sup> compared to conventional fixed-lifetime approaches. BrC DRE surpasses BC DRE in regions with significant biomass burning, underlining its important role in the regional radiation budget and its contribution to canceling out the negative DRE of OA.

In addition, the light absorption of BrC influences atmospheric chemistry by reducing the JNO<sub>2</sub> rate (0-7.4%), surface ozone concentrations (0 – 2.5%), and tropospheric OH concentrations (0 – 6.9%). The strongest effects were observed during intense wildfire events, with decreases of more than 36% in JNO<sub>2</sub> and more than 17% in ozone. In addition, BrC suppresses the enhancement of ozone and near-source OH from wildfire emissions, while amplifying global-scale OH reduction due to wildfire emissions, demonstrating its role in regulating atmospheric responses to wildfires.

# Code and data availability

The GEOS-Chem model is available at https://geoschem.github.io/ (last access: 15 February 2025). The code of version 12.8.2 can be downloaded at https://zenodo.org/records/3860693 (last access: 15 February 2025). The code for the

updated BrC simulation code is available at https://github.com/xiexinchun/xxc/tree/geoschem12.8.2-BRC (last access: 15 May 2025). All data can be obtained from the corresponding author upon request.

**Author contributions**. XCX and YZZ conceived and designed the study. XCX performed model development under the guidance of YZZ and XW. XCX performed all model simulations, data analyses, and visualization. RSL assisted with coding. XCX wrote the initial draft. YZZ and XCX revised the manuscript with contributions from all authors.

# Acknowledgement

This work was supported by the National Natural Science Foundation of China (42275112). The authors thank the High-Performance Computing Center of Westlake University and the National Supercomputing Center at Wuxi for facility support and technical assistance. XCX thanks Elijah Schnitzler for valuable discussion, which provided important insights for this study.

605

610

590

595

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
