# Peer review of "Global modeling of brown carbon: impact of temperature- and humidity-dependent bleaching"

_EGUsphere, 2025_

## Author Comment (AC1)

We thank the reviewer for constructive suggestions that help improve the manuscript. Below, we provide point-by-point responses, with reviewers' comments presented in black and our responses in blue.

**General Comments**

The manuscript by Xie et al. updates old assumptions about the reactivity of brown carbon (BrC), an important component of biomass burning emissions, by implementing a parameterization of its lifetime based on atmospheric conditions rather than just oxidant concentrations. They apply this parameterization to reassess BrC lifetime, its direct radiative effect (DRE), and its impacts on OH, $O_3$, and $NO_2$ photolysis. They also account for varying plume injection heights, improving upon previous studies (e.g., Schnitzler et al. 2022), which only modeled fast bleaching at low altitudes and no bleaching at high altitudes. Xie et al. use higher spatial resolution (altitude, longitude, and latitude), resulting in simulated BrC concentrations that better match observations. This is a significant advancement in understanding BrC's atmospheric role.

The paper is well written and important, but some statements are unclear, some arguments lack sufficient detail, and the limitations of the study should be discussed more thoroughly.

We thank the reviewer for recognizing the significance of this work.

**Specific Comments**

The limitations of the study should be discussed more thoroughly:

- The parameterization is based solely on reactions between $O_3$ and BrC, per Schnitzler et al. Other bleaching mechanisms should be acknowledged.
- The underlying study only considered BrC from smoldering pine wood, which may not represent atmospheric diversity.
- Only the water-soluble fraction of BrC was studied. The water-insoluble fraction, potentially more light-absorbing, was excluded.

Thanks for the suggestions. We have now added a Discussion section (Section 4) to discuss the limitations raised by the reviewer.

Section 4: "*While representing an important step forward, several aspects of the Schnitzler et al. (2022) parameterization merit further investigation. First, the experiment examined only ozone-driven bleaching. The role of other oxidants such as OH and their sensitivity to environmental parameters are yet to be explored. Second, the study focused on water-soluble BrC, though field evidence indicates that water-insoluble BrC accounts for a large fraction of BrC absorption. The impact of temperature and RH on bleaching behaviors of water-insoluble BrC remains uncharacterized. Third, the parameterization derives from BrC produced by smoldering pinewood, while real-world BrC properties may vary considerably across different fuel types and combustion conditions (Sun et al., 2021; Cai et al., 2023). Finally, neither the current study nor Schnitzler's work addresses secondary BrC*

*formation pathways, which may involve different bleaching mechanisms. These limitations highlight important directions for future research to further refine the representation of BrC transformation in atmospheric models.*"

**Line 166–168**: What is the reference for total OC, BC, and BrC emissions?

The emissions of OC and BC are based on a series of anthropogenic and natural emission inventories, which are described in Line 144-154 (Section 2.1). We now remove the sentence in Line 166-168 and state clearly in Line 155-158 that total OC and BC emissions are based on the sum of multiple emission inventories.

**Line 167–168**: "The total brown carbon emissions were 27.46 teragrams, of which biomass burning emissions were 21.17 teragrams." Since BrC is typically a subset of biomass burning emissions, this needs clarification. Please verify sources and define what is included in each total.

We now clarify in Line 154-158 Section 2.1 that "*These emission inventories yield global OC emissions of 42.74 Tg and BC emissions of 8.24 Tg in 2019. Of the total OC emissions, 21.17 Tg are from biomass burning and 6.29 Tg from biofuel burning. In the simulation, these sources are treated as BrC (see Section 2.2 for details), while the remaining OC emissions are modeled as purely scattering aerosols.*"

**Lines 215–231**: The aging discussion references Wang et al. (OH) and Schnitzler et al. ($O_3$). Their lifetimes may not be directly comparable. This should be noted.

We appreciate your comment and agree that the BrC lifetimes associated with OH oxidation and $O_3$ oxidation cannot be directly compared, since they represent different chemical aging mechanisms and environmental dependencies. In this work, for the purpose of modeling, we use Wang et al. or Schnitzler et al. to parameterize bleaching lifetime for all BrC, without explicitly accounting for chemical mechanisms. Now, following the reviewer's suggestion, we note both in the method description (Line 221-228 Section 2.2) and the newly added Discussion section (Section 4) that Schnitzler et al. experiment is based on $O_3$ oxidation and is for water soluble BrC only. We suggest that additional experiments are needed to improve on Schnitzler et al. parameterization.

Section 2.2: "*More recent laboratory experiments by Schnitzler et al. (2022) found that the chemical lifetime of water-soluble BrC, generated from smoldering pine wood in the presence of ozone, increases substantially with enhanced particle viscosity when temperature and RH get lower. Although the experimental setup (i.e., using ozone as the oxidant, generating aerosols from pine wood, and focusing on water-soluble BrC) does not fully represent atmospheric conditions, the finding that $\tau\_BrC$ is highly sensitive to environmental parameters has critical implications for BrC's radiative and chemical effects.*"

Section 4: "*While representing an important step forward, several aspects of the Schnitzler et al. (2022) parameterization merit further investigation. First, the*

*experiment examined only ozone-driven bleaching. The role of other oxidants such as OH and their sensitivity to environmental parameters are yet to be explored. Second, the study focused on water-soluble BrC, though field evidence indicates that water-insoluble BrC accounts for a large fraction of BrC absorption. The impact of temperature and RH on bleaching behaviors of water-insoluble BrC remains uncharacterized. Third, the parameterization derives from BrC produced by smoldering pinewood, while real-world BrC properties may vary considerably across different fuel types and combustion conditions (Sun et al., 2021; Cai et al., 2023). Finally, neither the current study nor Schnitzler's work addresses secondary BrC formation pathways, which may involve different bleaching mechanisms. These limitations highlight important directions for future research to further refine the representation of BrC transformation in atmospheric models.*"

**Line 469**: Figure S7 is described as showing the response to wildfire emissions (reactive gases + aerosols), but its caption says it shows only BrC absorption without biomass burning. These statements conflict.
- Clarify what is meant by "BrC absorption without biomass burning emissions." Is this just biofuel-derived BrC?

Thanks for pointing out this error. The caption now reads "*Figure S8. The effect of biomass burning emissions (including all gas and aerosol species) on global surface O3 (a), and column OH (b), computed as the difference between simulations with and without biomass burning emissions.*"

**Line 472**: References Figure S8, which does not exist. Without this figure, lines 473–476 cannot be evaluated.
Correction is made. It should be Figure S7 (Now as Figure S8).

**Technical Comments**
$JNO_2$ is used in the abstract without being defined.
The sentence in Abstract has been modified to "*Additionally, ...reducing photolysis rate of $NO_2$ ($JNO_2$) by up to 7.4%...*".

**Line 82**: Change "BrC around" to "BrC is around."
Thanks. The sentence has been changed as "*Studies have shown that the direct forcing effect (DRE) is around +0.1 W m$^{-2}$*".

**Line 86**: "What more" is awkward phrasing. Consider revising.
Now it has been changed to "*Moreover, ...*".

**Line 96**: "Tropopause" may be incorrect; did the authors mean "troposphere"?
Thanks. It has been changed to "troposphere".

**Line 243–246**: Three systems are said to be used, but only two (MC-LWCC and NOAA PILS-LWCC) are listed.

It should be "two systems". The correction is made.

**Line 279**: Change "In addition to 2019, but we also" to "In addition to 2019, we also" or simply "We also."
The sentence is modified as "*In addition to 2019, we also run ...*".

**Figure S4 caption**: Correct "ration of BrC" to "ratio of BrC."
This mistake has been fixed.

**Lines 474–476**: "The inclusion of BrC light absorption suppresses [...] near-source OH enhancement, but amplifies OH reduction [...]"
- The use of "but" implies a contradiction. Consider revising to: "...suppresses near-source OH enhancement and amplifies OH reduction..."
- Same revision suggested for lines 516–517 in the conclusion.

These two sentences have been revised to "*Comparison of Figure S8 and Figure 8 shows that including BrC light absorption partially offsets the ozone enhancement from wildfire emissions. Moreover, BrC absorption reduces near-source OH enhancement caused by fires while reinforcing OH reduction globally.*" Similar revision has been made for Line 580 in Conclusion.

**Lines 187–191**: Sentence is unclear. Revise for clarity.
Section 2.2 (Brown carbon simulation) has been re-written for clarity.

**Line 203**: Reference [50] appears where an author name is expected. Ensure consistent reference formatting.
This mistake has been fixed.

The term "fresh brown carbon" is used but not defined. Clarify that it refers to unbleached BrC if that is the intended meaning.
Thanks for the suggestion. We now clearly define fresh brown carbon and bleached brown carbon explicitly in Section 2.2: "*To capture the change of mass absorption efficiency (MAE) due to chemical processing, organic aerosols from biomass and biofuel burning are modeled as two distinct species: fresh BrC, which is strongly absorbing, and bleached BrC, which is weakly absorbing, following the approach by Wang et al. (2018). All freshly emitted OA from biomass and biofuel burning are specified as fresh BrC. They are subsequently converted to bleached BrC in the atmosphere, with the rate governed by bleaching lifetime $\tau_{BrC}$.*"

**Lines 375–378**: "The absorption of brown carbon becomes stronger with increasing altitude" is unclear. If referring to mass absorption efficiency, this needs clarification and justification.
This sentence is removed.

**Line 435**: The citation for Feng et al. appears to have the wrong year. Please verify

and correct.

This mistake has been fixed.

---

## Author Comment (AC2)

We thank the reviewer for constructive suggestions that help improve the manuscript. Below, we provide point-by-point responses, with reviewers' comments presented in black and our responses in blue.

Review of "Global modeling of brown carbon: impact of temperature- and humidity-dependent bleaching" by Xie et al.

Brown carbon (BrC) is a light-absorbing component of organic aerosols and is primarily emitted during biomass burning. BrC loses its light-absorbing capacity over time as it ages in the atmosphere by a process called bleaching. The study incorporates a new parameterization within GEOS-Chem model, based on a previous study (Schnitzler et al., 2022), that ties bleaching of BrC to environmental conditions (mainly temperature and humidity), instead of relying solely on oxidant concentrations. Using this approach, the study reassesses the impact of bleaching on BrC lifetime, atmospheric oxidants (OH, $O_3$), $NO_2$ photolysis and its direct radiative effect (DRE).

The study reports some key findings in that bleaching occurs much faster near the surface but becomes negligible at high altitudes. As a result, BrC lasts much longer in the atmosphere than previously thought, tripling global BrC levels and increasing its warming effect by 48%. The study also reveals that BrC absorption significantly reduces atmospheric chemistry activity, decreasing surface ozone by up to 2.5% and key atmospheric oxidants by up to 7%, with even stronger effects during major wildfire events.

The paper is mostly well written, and the study is important in improving our understanding of BrC burden, lifetime and its impacts on atmospheric oxidation and climate. However, some major revisions or clarifications are required for improving the analysis and discussions, especially model and observational comparisons. Further, overall limitations of the study should be better discussed throughout.
We thank the reviewer for recognizing the significance of this work.

Specific Comments:
1.  Line 83-86: Collow et al., (2024) is the not the first paper towards introducing BrC in the GOCART aerosol module within NASA GEOS Earth System Model. The study that first implemented BrC within GOCART to improve the simulated absorption in the near-UV was Colarco et al., (2017). Also, omit the phrase about "BrC e-folding time of 1 day" because that is not accurate either. The e-folding time in this case (within GOCART) has no inclusion of bleaching process, instead it is the rate of conversion of hydrophobic BrC to hydrophilic BrC, which in fact is 2.5 days based on Colarco et al. (2010). However, Collow et al. (2024) did evaluate the BrC implementation within GOCART on a global scale.
Thanks for the suggestion. The paragraph has been reorganized for clarity and rigor (Line 81-97 Section 1).

2. Line 126: "....parametrization in the model"-> Suggest introducing GEOS-Chem here instead of just saying "model", as well as clarify that it is a chemical transport model.

The sentence has been revised following the suggestion: "*In this study, we update the bleaching lifetime parameterization of BrC in the GEOS-Chem atmospheric chemical transport model, and examine the impacts of this update on the radiative effects of wildfire-derived BrC.*"

3. Line 132-136: Suggest combine the objectives of this study stated in these lines with the corresponding section numbers where the results from these objectives are discussed to make up a paragraph detailing how the manuscript is organized as follows.

Thanks for the suggestions. We have revised the text accordingly (Line 134-140 Section 1).

4. Lines 164-169: Please revise these sentences to explain/clarify what is being conveyed here? How is the relation between BrC/BC/OC/total BB emissions justify 65% emissions within PBL? What are the references for these emissions? As well as what BrC lifetime setting and experiments are being referred to here since "lifetime settings" and "sensitivity experiments" have not been discussed prior to this paragraph in the manuscript?

We have moved the relevant description to Line 292-300 Section 2.4 (Simulation experiments), after the lifetime setting and sensitivity experiments are presented. We now briefly explain the basis for the 65%:35% partition: "*To test the impact of vertical partitioning of fire emissions and its interactions with $\tau_{BrC}$ parameterizations, we perform GEOS-Chem simulations that assign 100% (0%) (Base and Upd) and 65% (35%) (BaFt and UpdFt) of wildfire emissions (all fire-emitted species including BC and BrC) in the boundary layer (free troposphere). The 65%:35% partitioning is based on the averages of aerosol smoke plume heights observed by the Multi-angle Imaging SpectroRadiometer (MISR) (Val Martin et al., 2010) and has been previously applied in GEOS-Chem modeling studies to assess the impact of fire plume heights (Fischer et al., 2014; Jin et al., 2023)*".

5. Lines 167: what fraction of OA is assumed to be fresh BrC emissions? What is the reference for the assumed emission factor for BRC? What is the source of Figure S3?

We have now rewritten Section 2.2 (Brown carbon simulation) for better clarity and logic, including explicit definition of "fresh BrC" and "bleached BrC" and their assumed properties. Briefly, we treat all freshly emitted biomass burning and biofuel OA as "fresh BrC", and "fresh BrC" is then converted to "bleached BrC" (with 1/4 light absorption of fresh BrC) at a rate governed by bleaching lifetime $\tau_{BrC}$. Biomass burning emissions are from GFED4s and biofuel emissions from Bond et al. We now add this information to the text and the caption of Figure S3.

6. Lines 286/Section 3.1: Clarify within this section or earlier that Chemical lifetime ($t_{BRC}$) referred to throughout this study only accounts for the bleaching process. There can be several other processes than can impact BRC chemical lifetime such as "browning of BRC" due to functionalization of BrC compounds (DeLessio et al., 2024; Schnitzler et al., 2020).

Thanks for the suggestion. We now use "bleaching lifetime" throughout the manuscript. This is formally defined in Section 2.2: "*The conversion from fresh to bleached BrC is governed by the chemical lifetime $\tau_{BrC}$ (referred to as bleaching lifetime hereafter).*"

7. Section 3.2: How is the fresh and aged BRC tracked in the model? It is not explained either in model description or this section.

We have now rewritten Section 2.2 (Brown carbon simulation) to clarify how we treat fresh and bleached BrC in our model. Briefly, we model organic aerosols originated from biomass and biofuel burning as two distinct species: fresh BrC (strongly absorbing) and bleached BrC (weakly absorbing), following the approach by Wang et al. (2018). All freshly emitted OA from biomass and biofuel burning are specified as fresh BrC. They are subsequently converted to bleached BrC in the atmosphere, with the rate governed by bleaching lifetime $\tau_{BrC}$. In this study, we investigate the impact of an updated $\tau_{BrC}$ parameterization, as a function of temperature and relative humidity, on BrC simulation.

8. Figure 2: Please add the area-weighted global mean values at the top each of the panels for the respective quantities. Also, please add a row with total BRC column density UpdFt and difference between total BRC UpdFt and BaFt.

Thank you for your suggestion. We have added a row with total BrC column in Fig. 2. The area-weighted means of fresh BrC for global, tropics, northern extratropics, and southern extratropics are shown in Fig. 3c. We now denote Fig. 3c with numerical values to help readers compare.

9. Line 351-53: What is the baseline simulation referred to here, BaFt? If so, please clarify this here because the differences in BRC column load are due to two factors here: (1) changes in $t_{BRC}$ and (2) changes in injection altitudes, and it should be clear to the readers that tripling of BRC column burden is primarily due to which factor. Accordingly change the language in the abstract and conclusion.

We now explicitly state here and elsewhere (including abstract and conclusion) which simulation is used for the discussion. For instance, in Section 3.2: "*Globally, the updated simulation (**UpdFt**) produces an average column density of 144.4 μg m$^{-2}$, about three times that of the baseline simulation (**Base**, 53.0 μg m$^{-2}$)... Comparison with **BaFt and Upd** simulations (average column density of 52.9 μg m$^{-2}$ and 120.7 μg m$^{-2}$) suggests that the increase in BrC abundance in UpdFt relative to Base is mainly due to the bleaching lifetime update.*"

We also clarify, through comparison between Base, BaFt, Upd, and UpdFt simulations, that tripling of BrC column burden is mainly due to updated tau_BrC (Base, Upd, and UpdFt); however, injection heights have a secondary, interactive effect because of the environmental condition dependent $\tau_{BrC}$ (Upd and UpdFt). This effect is not seen in original $\tau_{BrC}$ parameterization which is insensitive to environmental conditions (Base and BaFt). We have revised Section 3.2 (Global distribution of fresh BrC) to more logically present and discuss the above points.

10. Lines 391-410: The discussion here infers that the underestimation of model AAOD is not compensated by increasing BrC load rather appears to be stemming from either assumptions of BRC/OA emission ratio or the assumptions of BrC microphysical and optical properties, especially size distribution and refractive index assumptions that need further adjusting using observations. It is worth adding this to the discussions here.

Thank you for your insightful comments. We note during this revision that our original comparison of AERONET data and the model did not include data filtering, resulting in inconsistent model-observation comparison. Level 2.0 AERONET data are only available for a subset of days due to multiple reasons. We now sample the model simulations on days and locations when AERONET data is available (previously monthly means at each station are used). This procedure is now explained Section 2.3 (Observational data). The updated results are shown in Figures 5 and 6, which generally shows reduced discrepancy between the simulated and observed AOD and AAOD.

Following the suggestion, we also add discussion on the uncertainties associated with biomass burning emissions and BrC optical properties in Section 3.3:

"*The underestimation of Abs365 against ATom-4 data (Figure 4) and AAOD against AERONET data (Figure 5 and 6) may partly be explained by underestimation of biomass burning emissions. We perform additional simulations with the newly released GFED5 fire emission inventory (https://globalfiredata.org/, Last access: 10 April, 2025), which generally predicts higher biomass burning emissions than the GFED4s inventory. This simulation leads to better alignment with both ATOM-4 Abs365 (Figure 4) and AERONET AAOD observations (Figure 6). However, the GFED5 simulatoin also leads to an overestimation of OA in the lower troposphere against ATom-4 data and AOD against AERONET data (Figure 4 and 6).*"

"*Additionally, there are also considerable uncertainties in the BrC simulation associated with its optical properties. The MAE applied in different modeling studies vary considerably (Zhang et al., 2020; Jo et al., 2016), and laboratory measurements have also demonstrated source- and season-dependent differences in MAE (Chen et al., 2018; Xie et al., 2020). Moreover, assumptions about particle size distribution and refractive index also contribute to the uncertainties and need to be further constrained using observational and experimental data (Wu et al., 2020; Shamjad et al., 2018).*"

*Future improvement of BrC simulations may including refined treatment of these factors.*"

11. In addition to AOD and AAOD comparisons, SSA comparisons (with AERONET observations) would also be useful to investigate further on the model assumptions of BRC optical properties.

We now add Figure S5, which compares simulated and observed SSA. The simulated SSA values are generally higher, consistent with the underestimation of AAOD. Also, Figure 4b shows that the model generally overestimates total organic aerosol. In addition to underestimated BrC absorption, the overestimation of scattering OA may also contribute to higher SSA.

12. Section 3.3: Is Fig. 5 annual mean AOD/AAOD plots? If yes, it should be mentioned so in the caption and in the discussion.

Yes, the results in Figure 5 are the annual mean AOD and AAOD. We now mention this in the caption and in the discussion.

13. Lines 401-403: I don't think AOD threshold for AAOD retrievals explains the model underestimation of AAOD at all. Was the model data not filtered using the same threshold? If not, Fig 6b should include such filtering for a credible comparison and to rule out such a bias. Since a full year of model data is used for mean AAOD in Fig. 6b, there should still be a good enough sample size left after filtering.

We now eliminate this factor by applying the same data filtering for observations and simulations, so the comparison is apple-to-apple. The updated results show improved correlation between simulations observations but still could not fully account for the remaining discrepancies. Additional description and discussion have been included accordingly (see response to Comment No. 10).

14. 6 and Section 3.4: It is hard to reconcile that even though model AOD and AAOD (less so) did not change much between UpdFt and BaFt compared to AERONET observations (since r and slope are effectively the same for both experiments), how come there is such a big impact on DRE estimations (i.e. a change of 48% in DRE between the two experiments)? Can this be better explained?

Thanks for pointing this out.

First, we have corrected an error in our program to generate the previous AAOD plot. With the corrected Fig.6, the difference in AAOD between UpdFt and BaFt is more pronounced. On the other hand, the small change in AOD is expected. As AOD is dominated by aerosol scattering, updating $\tau_{BrC}$ leads to only negligible changes in AOD.

Second, AAOD accounts for absorption by all light-absorbing aerosols, including BC, BrC, and dust, whereas the reported DRE values only consider absorption by BrC.

Since the sole difference between the UpdFt and BaFt simulations is the BrC bleaching lifetime, the relative changes in DRE are naturally more pronounced than those in AAOD when comparing the two simulations.

Finally, the AERONET sites do not cover remote regions (e.g., remote oceans) where the largest relative changes in BrC absorption occur (Reviewer Response Figure 1).

[Figure]

Reviewer Response Figure 1. The difference DRE of BrC from UpdFt and BaFt.

**Editorial Comments:**
Line 86: replace 'What more' by 'Furthermore'.
Changes have been made following the suggestion.

Line 166: replace 'coincident' with 'consistent'.
This section has been rewritten. The sentence is now removed.

Line 361: replace "...the $t_{BRC}$ parameterization will not change" with "the updated $t_{BRC}$ parameterization did not change"
The sentence "…the $\tau_{BrC}$ parameterization will not change…" has been replaced to "Meanwhile, the updated $\tau_{BrC}$ parameterization does not change the total BrC concentrations…"

Line 202: BC/OA [50], what is 50 here? Is it a reference? Please add a legit reference here.
The reference is now cited correctly.

**Technical Comment:**
"Data availability" section is missing at the end of the manuscript. There is no statement (or a DOI) regarding accessing of modeling and observational data used in this study. At the very least, modeling data used to produce the figures in the manuscript should be made publicly available.
Thanks for the suggestion. We have added this part as follows:

***Code and data availability***
*The GEOS-Chem model can be available at https://geoschem.github.io/ (last access: 15 February 2025). The code of version 12.8.2 can be downloaded at*

*https://zenodo.org/records/3860693 (last access: 15 February 2025). The updated code with BrC simulation is available from GitHub: https://github.com/xiexinchun/xxc/tree/geoschem12.8.2-BRC (last access: 15 May 2025). All data can be obtained from the corresponding author upon request.*

**References:**

Colarco, P. R., Gassó, S., Ahn, C., Buchard, V., da Silva, A. M., and Torres, O.: Simulation of the Ozone Monitoring Instrument aerosol index using the NASA Goddard Earth Observing System aerosol reanalysis products, Atmos. Meas. Tech., 10, 4121–4134, https://doi.org/10.5194/amt-10-4121-2017, 2017.

Colarco, P., da Silva, A., Chin, M., and Diehl, T.: Online simulations of global aerosol distributions in the NASA GEOS-4 model and comparisons to satellite and ground-based aerosol optical depth, J. Geophys. Res.-Atmos., 115, D14207, https://doi.org/10.1029/2009JD012820, 2010.

DeLessio, M. A., Tsigaridis, K., Bauer, S. E., Chowdhary, J., and Schuster, G. L.: Modeling atmospheric brown carbon in the GISS ModelE Earth system model, Atmos. Chem. Phys., 24, 6275–6304, https://doi.org/10.5194/acp-24-6275-2024, 2024.

Schnitzler, E. G., Liu, T., Hems, R. F., and Abbatt, J. P. D.: Emerging investigator series: heterogeneous OH oxidation of primary brown carbon aerosol: effects of relative humidity and volatility, Environ. Sci.-Proc. Imp., 22, 2162–2171, https://doi.org/10.1039/D0EM00311E, 2020.